# Discovering Intrinsic Spatial-Temporal Logic Rules to Explain Human Actions

**Chengzhi Cao**[1,2]*, **Chao Yang**[1], **Ruimao Zhang**[1], **Shuang Li**[1†]
[1]The Chinese University of Hong Kong (Shenzhen)
[2]University of Science and Technology of China
`chengzhicao@mail.ustc.edu.cn, 222043011@link.cuhk.edu.cn,`
`zhangruimao@cuhk.edu.cn, lishuang@cuhk.edu.cn`

## Abstract

We propose a *logic-informed* knowledge-driven modeling framework for human movements by analyzing their trajectories. Our approach is inspired by the fact that human actions are usually driven by their intentions or desires, and are influenced by environmental factors such as the spatial relationships with surrounding objects. In this paper, we introduce a set of *spatial-temporal logic rules* as knowledge to explain human actions. These rules will be automatically discovered from observational data. To learn the model parameters and the rule content, we design an expectation-maximization (EM) algorithm, which treats the rule content as latent variables. The EM algorithm alternates between the E-step and M-step: in the E-step, the posterior distribution over the latent rule content is evaluated; in the M-step, the rule generator and model parameters are jointly optimized by maximizing the current expected log-likelihood. Our model may have a wide range of applications in areas such as sports analytics, robotics, and autonomous cars, where understanding human movements are essential. We demonstrate the model's superior *interpretability* and *prediction* performance on pedestrian and NBA basketball player datasets, both achieving promising results.

## 1 Introduction

For a human, although the exhibited movements can be complex, the logic behind the actions is usually simple, clear, and can be generalized [17]. The *logic rules* present a compact and high-level *knowledge representation*, defining what actions tend to be executed under what conditions. It emphasizes reasoning, and encourages structuring the world in terms of objects, properties, and relations [22]. There has been a great interest and business value in unveiling the human logic from the observational movements and actions [18]. We provide two motivating examples below.

In *sports analytics*, understanding each player's behavior preferences or tendencies under various scenarios will provide coaches with valuable information [1]. Usually, coaches need to watch the game or training videos for hundreds of hours before they can summarize the discoveries into compact principles. *Could we design an algorithm to synthesize these principles from the raw action data automatically*? One can imagine, such a tool will significantly reduce the workload of coaches, providing more granular insight into each player's capabilities and strategies, and aiding in personalized training and match strategy design [15].

In *self-driving car*, it's essential to enable the self-driving cars to "read human's mind" like humans. This requires the self-driving cars to automatically *understand human intentions and reasoning* when

---

*Work done as a graduate research assistant at The Chinese University of Hong Kong (Shenzhen)
†Corresponding author.

they are running on the same roads with human drivers [2]. If self-driving cars can automatically distill logic rules from their observed *low-level noisy* human actions and movement trajectories, it will increase the technical reliability and accelerate the widespread use of self-driving cars.

For human actions, lots of the governing rules would be regarding the *spatial-temporal relation* with the surrounding environments and their intentions [9]. For example, when a basketball player with a ball is within the scoring range, his/her action, such as shoot, pass, or triple threat, is influenced by historical and current surrounding factors, such as the current locations of the player and the defenders, the time elapses of the game, the success shooting rate of the player in today's game, and so on. The quick decision made by the player is actually reflecting a composition of all these factors, which can be described as a collection of spatial-temporal logic rules in our model.

Formally, in our introduced spatial-temporal logic rules, the logic variable (i.e., predicate) set will include spatial-temporal relation predicates, in addition to the commonly defined object property and relation predicates. The rule content will *capture the spatial relation of the object with surrounding objects, as well as the temporal ordering constraints of the events*.

Our methods have the following distinct features:

*From the modeling perspective:* (*i*) Our human action model is a rule-based *probabilistic* model, which treats each hidden rule as a "soft" constraint. We assume each rule will be executed by humans with probabilities, and this *tolerates the uncertainties* in data. (*ii*) Our model directly uses low-level, fine-grained, and (may) irregularly-spaced action times and locations (i.e., original *3d coordinates*) as inputs, as opposed to other rule-based models, where one needs to first extract relational data as inputs. (*iii*) Our spatial and temporal predicates are also probabilistic. For predicates such as "left of" or "before", we model them as kernel functions with learnable variables. In this way, our introduced spatial-temporal predicates are *smooth functions* of the input locations and times, which increases model flexibility.

*From the learning perspective:* We propose a *tractable* and *differentiable* algorithm that can jointly learn the *rule content* and *model parameters* from observational data. The learning framework is designed to maximize the likelihood of the human action trajectories. Specifically, we propose to use a neural rule generator to generate the spatial-temporal logic rule set. Our continuous rule generator parameters will be optimized in a differentiable way. The overall learning procedure is an expectation-maximization (EM) algorithm, where we treat the rule set as latent variables. In the E-step, the posterior distribution over the latent rule set is evaluated. In the M-step, both the rule generator parameters and the model parameters are optimized by maximizing the expected log-likelihood with respect to the current posterior. We demonstrated the promising performance of our model in terms of human action *prediction* and *explanation* on two interesting real datasets.

## 2   Related Work

**Logic Rule Learning.**   Learning logic rules from raw data has been widely studied for various downstream tasks, such as motion inference [4] and healthcare analysis [11]. Learning rules via an exact search requires enumerating all combinations of the logic predicates and is intractable in most problems. One has to design heuristic searching algorithms by leveraging some structural properties of the problems. For example, Dash et al. [5] formulated a convex rule learning problem and proposed a column generation algorithm to expand the rule set gradually. Wang et al. [25] designed a Bayesian framework for learning rule classifiers and derived bounds on the support of rules in a MAP solution. Recently, Yang et al. [28] proposed an interesting end-to-end differentiable approach (Neural LP) to learn the parameters and structure of logical rules. Qu et al. [19], and Sadeghian et al. [22] proposed an efficient logic rule mining algorithm based on the knowledge graph data. *However, none of these advanced rule mining methods can directly work on spatial-temporal human action data when the inputs are raw event 3d coordinates and types.*

**Spatio-Temporal Dynamics for Event Data.**   Since the human actions are irregular spatial-temporal event data, we also briefly discuss *probabilistic* models for such *event sequences*. Modeling the spatial-temporal dynamics of discrete events is foundational in many scientific fields and applications [20]. Shen et al. [23] proposed a novel deep learning model for spatial-temporal events such as taxi data and achieved promising prediction accuracy. Zhou et al. [31] integrated deep learning methods with spatiotemporal point processes and modeled the intensity function as a latent stochastic process. Chen et al. [3] deployed two novel architectures, including jump and attentive continuous-time normalizing flows, to learn the dynamics of the spatiotemporal event data. Repe et al.

[21] learned canonical spatiotemporal point cloud representation using a latent ODE and continuous normalizing flows to generate shapes continuously in spacetime. *However, these spatial-temporal event models are governed by hard-to-interpret dynamic functions and cannot be generalized to model human action events. Could we propose a model with logic-informed dynamic functions to explain the spatial-temporal human action events?*

## 3 Our Model

### 3.1 Data: Human Actions Recorded as Spatial-Temporal Event Sequences

Consider a set of objects, denoted as $\mathcal{C}$. For the object $c \in \mathcal{C}$, its trajectories and the key actions observed up to $t$ can be summarized as a sequence of temporally ordered events:

$$\mathcal{H}_{t-}^c = \{e_1^c = (t_1^c, s_1^c, \kappa_1^c), \ldots, e_n^c = (t_n^c, s_n^c, \kappa_n^c) \mid t_n^c < t\}, \tag{1}$$

where $t \in R^+$ is the time, $s \in R^2$ is the location, and $\kappa \in \mathcal{K}$ is the event (i.e., action) type.

### 3.2 Definition of Spatial-Temporal Predicates

**Static Predicate** Given the object set $\mathcal{C}$, the *predicate* is defined as the *property* or *relation* of objects, which is a *logic function* as follows:

$$X(\cdot) : \mathcal{C} \times \mathcal{C} \cdots \times \mathcal{C} \mapsto \{0, 1\}. \tag{2}$$

For example, $Smokes(c)$ is a property predicate and $Friend(c, c')$ is a relation predicate.

**Temporal relation predicates** They can be used to define the temporal relations of two action events. We consider three types of temporal relation predicates {before, equal, after} as:

$$\begin{aligned} R_{\text{Before}}(t_1, t_2) &= \mathbb{1}\{t_1 - t_2 < 0\}, \\ R_{\text{After}}(t_1, t_2) &= \mathbb{1}\{t_1 - t_2 > 0\}, \\ R_{\text{Equal}}(t_1, t_2) &= \mathbb{1}\{t_1 = t_2\}. \end{aligned} \tag{3}$$

We will treat the temporal relation predicate as either a boolean variable or a real-valued function. If the time information is imprecisely recorded, we can parameterize the temporal relation predicates as temporal kernel functions that map to $[0, 1]$, which is a function of $t_1, t_2$ with learnable parameters.

**Spatial-Temporal Predicate** In our paper, we extend the above static predicates to spatial-temporal predicates, which include spatial-temporal *property* predicates and spatial-temporal *relation* predicates.

Specifically, the spatial-temporal *property* predicates are defined as:

$$X(\cdot) : \mathcal{C} \times \cdots \times \mathcal{C} \times \mathcal{T} \times \mathcal{S} \mapsto \{0, 1\}. \tag{4}$$

For example, $PickupKey(c, t, s)$ is a spatial-temporal property predicate. Suppose an entity $c_1$ picked up the key at time $t_1$ in location $s_1$, then the predicate will be grounded as True (1) at $(c_1, t_1, s_1)$, i.e., $PickupKey(c_1, t_1, s_1) = 1$; otherwise it is false.

Given the observational human action data, the grounded predicate $\{PickupKey(c, t, s)\}_{t=1,2,\ldots}$ can be modeled as a sequence of discrete events – when the predicate becomes True, an event happens. In general, the grounded spatial-temporal property predicate is a discrete event sequence, where the event occurrence times and locations are irregular.

The spatial-temporal *relation* predicates are introduced to define the spatial and temporal relations of two entities. Specifically, they are defined as:

$$R(\cdot, \cdot) : (\mathcal{C} \times \mathcal{T} \times \mathcal{S}) \times (\mathcal{C} \times \mathcal{T} \times \mathcal{S}) \mapsto \{0, 1\}. \tag{5}$$

Spatial-temporal relation predicates are logic variables indicating the spatial-temporal relations of two objects, where we further divide them into *temporal relation* predicates, *static spatial relation* predicates, and *dynamic spatial relation* predicates. More details can be found in Appendix.

It is noteworthy that all these boolean predicates can be converted to probabilistic ones. We can soften these logic functions by kernel functions with learnable parameters to tolerate uncertainties in data.

### 3.3 Definition of Spatial-Temporal Logic Rules

We will consider spatial-temporal logic rules where the body part contain spatial-temporal predicates as relation constraints. For example, a sensible rule will look like:

$$\begin{aligned} f : Y_{\text{TurnAround}}(c, t, s) &\leftarrow X_{\text{PickUpKey}}(c, t, s) \bigwedge \\ R_{\text{InFront}}((c', t, s'), (c, t, s)) &\bigwedge R_{\text{Behind}}((c'', t, s''), (c, t, s)), \end{aligned} \tag{6}$$

where $c \in \mathcal{C}_{\text{person}}$, $c' \in \mathcal{C}_{\text{block}}$, and $c'' \in \mathcal{C}_{\text{key}}$. "person", "block" and "key" are the specific instances in the object set $\mathcal{C}$, and this rule represents that a person wants to pick up a key, while one block is in front of him and the key is behind him, so he turns around. We utilize this example to manifest the meaning of logic rules. In general, the *spatial-temporal logic rule* in our paper is defined as a logical connectives of predicates, including property predicates and spatial-temporal relation predicates:

$$f : Y(v) \leftarrow \bigwedge_{X_{\text{property}} \in \mathcal{X}_f} X_{\text{property}}(v) \bigwedge_{R_{\text{spatial-temporal}} \in \mathcal{R}_f} R_{\text{spatial-temporal}}(v', v), \tag{7}$$

where $Y(v)$ is the *head predicate* evaluated at the entity-time-location triplet $v$, $\mathcal{X}_f$ is the set of property predicates defined in rule $f$, and $\mathcal{R}_f$ denotes the set of spatial-temporal relation predicates defined in rule $f$.

### 3.4 Logic-Informed Action Event Models

We consider a setting where we can fully observe the trajectories of all the moving objects, including their real-time locations and key actions (i.e., events), denoted as $\mathcal{H}_t$. We aim to propose a logic-informed spatial-temporal model to predict and explain the action type given the entity-time-location triplet $v = (c, t, s)$ (i.e., query) and $\mathcal{H}_t$. The main idea is to construct the model features using spatial-temporal logic rules, as defined in Eq. (7). Intuitively, given the entire trajectories $\mathcal{H}_t$ and the query $v = (c, t, s)$, the body part of the rule defines the evidence to be selectively gathered from *history* to deduce the event type for query entity $v = (c, t, s)$. Assume that for each possible event type $\kappa \in \mathcal{K}$, there exist multiple rules such as Eq. (7) to explain its occurrence, with $\kappa$ being the *head predicate*. Given an individual rule as Eq. (7), we propose to build the *feature* that is conditional on history and query as

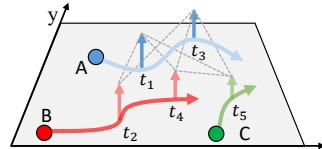

Figure 1: Illustration of feature construction using a simple logic formula with temporal relation predicate $(t_1 < t_2)$, $f : Y \leftarrow A \wedge B \wedge C \wedge (A \text{ Before } B)$. The rule defines the template to gather combinations of the body predicate history events. Here predicate A has 2 events and predicate B has 1 event, the temporal relation constraint would lead to valid combinations (also called "paths"). This type of feature construction can be extended to spatial-temporal cases, where we count the valid paths as the feature.

$$\phi_f(\kappa | v, \mathcal{H}_t) = \text{sign}(\kappa \in f) \cdot \sum_{\text{path} \in \{\mathcal{H}_t, v\}} g_f(\text{path}), \tag{8}$$

where we introduce a function $g_f(\cdot)$ to check the body conditions of $f$ given a "path". We use a simple example to explain how to compute features, as shown in Figure 1. As illustrated, the feature computes the valid total number of "paths" given the data and query.

Suppose there is a rule set $\mathcal{F}_\kappa$, where the event $\kappa$ is the head predicate. All the rules will play together to reason about the occurrence of $\kappa$. For each $f \in \mathcal{F}_\kappa$, one can compute the features as above. Given the rule set $\mathcal{F}_\kappa$, we model the probability of the event $\kappa$ as a log-linear function of the features, i.e.,

$$p(\kappa | v, \mathcal{H}_t) \propto \exp\left(\sum_{f \in \mathcal{F}_\kappa} w_f \cdot \phi_f(\kappa | v, \mathcal{H}_t)\right), \tag{9}$$

where $w = [w_f]_{f \in \mathcal{F}} \geq 0$ are the learnable weight parameters associated with each rule. All the model parameters can be learned by maximizing the likelihood, which can be computed using the above Eq. (9). We intend to train a rule generator $p_\theta$ and an evaluator $p_w$ to maximize the likelihood of training data as:

$$\max_{\theta, w} \mathcal{O}(\theta, w) = \mathbb{E}_{(\kappa, v, \mathcal{H}_t)}[\log \mathbb{E}_{p_\theta}[p_w(\kappa | v, \mathcal{H}_t)]]. \tag{10}$$

More details can be found as follows.

## 4 Our Learning Algorithm

Our goal is to jointly learn the set of spatial-temporal logic rules $\{\mathcal{F}_\kappa\}_{\kappa \in \mathcal{K}}$ and their weights by the maximum likelihood method, where each rule has a general form as Eq. (7).

To discover each rule, the algorithm needs to navigate through the combinatorial space considering all the combinations of the property predicates and their spatial and temporal relations. To address this computational challenge, we propose a tractable (functional) EM algorithm that treats the rule set as latent variable $z$. The rules will be generated by a hidden neural rule generator. The overall learning framework alternates between an E-step, where the posterior distribution of the latent rule space is evaluated (rule generation), and the M-step, where the model parameters and rule generator parameters are optimized. Please refer to Figure. 2 for an illustration.

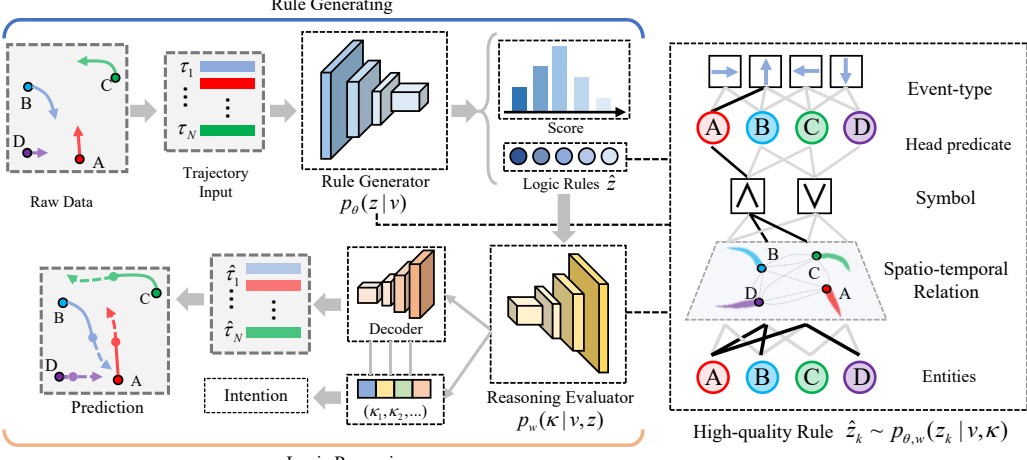

Figure 2: The overview of our proposed framework. It contains two important processes: rule generating and logic reasoning. Given the past motion of each entity on a scene over the last few seconds, the rule generator generates logic rules for the reasoning predictor. The reasoning predictor takes the generated rules as input, and predict the intention of each entity. It is optimized by EM algorithm. In the E-step, a set of top K rules are selected from all generated rules via posterior inference. Finally in the M-step, the rule generator is updated to be consistent with the high-quality rules identified in E-step.

Our goal is to maximize the likelihood of the observed human action events $\{\kappa^{(i)}\}_{i=1,...,n}$. Using the chain rule, we have

$$\log p_w(\{\kappa^{(i)}\}_{i=1,...,n}) = \sum_{i=1}^{n} \log p_w(\kappa^{(i)} \mid v^{(i)}, \mathcal{H}_{t(i-1)}). \tag{11}$$

To simplify the notation, we will use $p_w(\kappa^{(i)})$ to stand for $p_w(\kappa^{(i)} \mid v^{(i)}, \mathcal{H}_{t(i-1)})$ in the following. Given a latent rule set $z$, we have to marginalize the posterior of $z$ to get the above log-likelihood. However, the exact inference of $z$ is intractable. We will introduce an amortized recognition network $p_\theta(z|\kappa^{(i)})$ to approximate the true posterior. We have

$$\log p_w(\kappa^{(i)}) = D_{KL}(p_\theta(z|\kappa^{(i)})||p_w(z|\kappa^{(i)})) + \mathcal{L}(\theta, w; \kappa^{(i)}), \tag{12}$$

where the first term is the KL divergence of the approximate from the true posterior, and the second term $\mathcal{L}(\theta, w; \kappa^{(i)})$ is the variational lower bound (ELBO). It can be represented as:

$$\mathcal{L}(\theta, w; \kappa^{(i)}) = -D_{KL}(p_\theta(z|\kappa^{(i)})||p_w(z)) + \mathbb{E}_{p_\theta(z|\kappa^{(i)})}[\log p_w(\kappa^{(i)}|z)]. \tag{13}$$

And $\log p_w(\kappa^{(i)}) \geq \mathcal{L}(\theta, w; \kappa^{(i)})$. The bound becomes tight when the approximate posterior matches the true one. Our goal is to optimize the variational parameters $\theta$ and model parameters $w$ from the ELBO lower bound.

## 4.1 Rule Generator

We deploy Transformer-based framework to model the rule generator $p_\theta$. We define the distribution of a set of rules as follows:

$$p_\theta(z \mid v, \mathcal{H}_t) = \Psi(z|N, \text{Trans}_\theta(v, \mathcal{H}_t)), \tag{14}$$

where $\Psi(\cdot)$ is multinomial distributions, $N$ is the number of the top rules, and $\text{Trans}_\theta(v, \mathcal{H}_t)$ defines a distribution over compositional rules with spatial-temporal states. The generative process of the rule set is quite intuitive, where we simply generate $N$ rules to form $z$. In fact, this $p_\theta(z \mid v, \mathcal{H}_t)$ is a flexible posterior approximation function, which will be optimized by the EM type algorithm.

We choose transformer over graph neural network (GNN) as our baseline because transformer architectures are based on a self-attention mechanism that is able to capture long-range relationships, as opposed to recurrent neural networks that process sequence elements recursively and can only take into account short-term context. Note that the graph operations in GNN are designed to learn node representations on the fixed and homogeneous graphs. The limitations especially become problematic when learning representations on a changeable graph that consists of various types of nodes and edges.

## 4.2 Rule Evaluator

Eq. (9) is our rule evaluator (suppose we know the rule content). Here we assume the rule content is latent, and the rule evaluator is given as

$$p_w(\kappa|v, z, \mathcal{H}_t) = \frac{\exp\left(\sum_{f \in z_\kappa} w_f \cdot \phi_f(\kappa|v, \mathcal{H}_t)\right)}{\sum_{\kappa'} \exp\left(\sum_{f \in z_{\kappa'}} w_f \cdot \phi_f(\kappa'|v, \mathcal{H}_t)\right)}. \tag{15}$$

## 4.3 Optimization

We optimize the rule generator $p_\theta$ and reasoning evaluator $p_w$ to maximize the objective in Eq. (10). At each training iteration, we first update the reasoning predictor $p_w$ according to some rules generated by the generator, and then update the rule generator $p_\theta$.

In our network, the latent rule set will be automatically discovered. The best set of logic rules is approximately obtained by sampling and preserving the top-K rules according to their posterior probabilities. Specifically, as shown in Eq. (14), the posterior probabilities of the latent rule $z$ is obtained by a Transformer type of encoder, which maps the input observed action trajectories to a latent explanatory rule space. Each candidate rule is generated in the latent rule space token-by-token (token means logic variable/predicate in our setting) in a sequential manner and meanwhile the posterior probability of each rule sequence can be evaluated. When optimizing the evaluator, we draw several rules $\hat{z}$ for each query and let the evaluator uses $\hat{z}$ to predict $\kappa$. For each query, we try to identify top-K rules $z_I$ from all generated rules $\hat{z}$. It is accomplished by taking into account the posterior probabilities of each subset of logic rules $z_I$ with prior from the rule generator $p_\theta$ and likelihood from the reasoning predictor $p_w$. Specifically, when a series of rules produced from the rule generator $p_\theta$, we calculate the weights of each rule $z^{(i)}$ as follows:

$$J(z^{(i)}) = \{p_w(\kappa|z^{(i)}) - \frac{1}{|\mathcal{A}|}\} + \log \text{Trans}_\theta(z^{(i)}|v, \mathcal{H}_t), \tag{16}$$

where $\mathcal{A}$ is the set of all candidate event type inferred by logic rules. $\text{Trans}_\theta(z^{(i)}|v, \mathcal{H}_t)$ is the probability of rule computed by the generator. For a subset of rules $z_I \subset \hat{z}$, the log-probability can be approximated as: $\log p_{\theta,w}(z_I|v, \mathcal{H}_t) \approx \sum_{z^{(i)} \in z_I} J(z^{(i)}) + \log \Psi(z_I|N, \text{Trans}_\theta(v, \mathcal{H}_t)) + \text{const}$. This equation inspired us to use the distribution $q(z_I) \propto \exp(\sum_{z^{(i)} \in z_I} J(z^{(i)}) + \log \Psi(z_I|N, \text{Trans}_\theta(v, \mathcal{H}_t)))$ as approximation of the posterior. Each rule $z^{(i)}$ sampled from $q(z_I)$ independently can be formed with $N$ logic rules.

Clearly, $J(z^{(i)})$ can be regarded as the quality of candidate rules, with consideration of the evaluator $p_w$. It is calculated as the contribution of a rule to the correct event type minus the average contribution of this rule to the other candidate responses. A rule is more significant if it obtains a higher score to the correct event type and a lower score to other potential predictions.

After getting several high-quality rules from training data, we further utilize these rules to update the parameters of rule generator $p_\theta$. Concretely, we regard the generated high-quality rules as part of training data, and update the rule generator by maximizing the log-likelihood as follows:

$$\mathcal{O}(\theta) = \log p_\theta(z_I|v, \mathcal{H}_t) = \sum_{z^{(i)} \in z_I} \log \text{Trans}_\theta(v, \mathcal{H}_t) + \text{const}. \tag{17}$$

By learning to generate high-quality rules, the rule generator will reduce the search area and produce better empirical results for the reasoning predictor.

## 5 Experiments

In this section, we provide some implementation details and show ablation studies as well as visualization to evaluate the performance of our framework. We compare our model with several state-of-the-art approaches, including PECNet [14], NMMP [7], STGAT [8], SOPHIE [22], STAR [29], Y-Net [13], MID [6], Social-SSL [24], Social-VAE [26], Social-Implicit [16], and NSP-SFM [30].

### 5.1 Implementation

We follow the same data prepossessing strategy as PECNet [14] for our method. All models were trained and tested on the same split of the dataset, as suggested by the benchmark. We train the network using Adam optimizer with a learning rate of 0.001 and batch size 16 for 500 epochs.

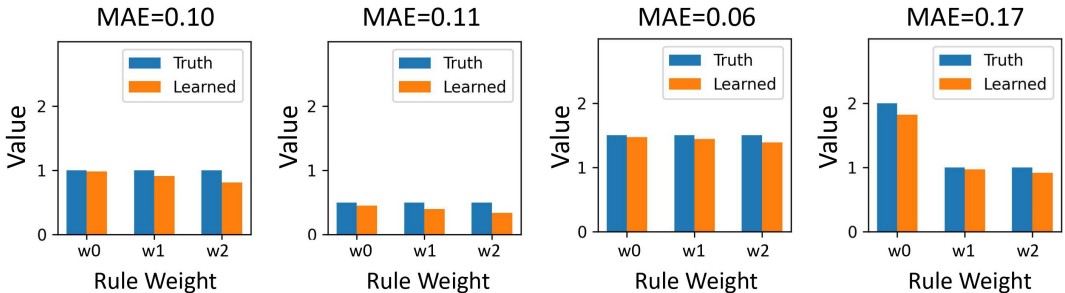

Figure 3: Rule discovery and weight learning results on 4 synthetic datasets (2.4K seqs).

Table 1: Quantitative results ($ADE_{20}/FDE_{20}$, and accuracy) of trajectory prediction in NBA dataset. The bold/underlined font represent the best/second best result.

| Times | Y-Net | MID | NSP-SFM | Social-SSL | Social Implicit | Social VAE | Ours |
|-------|-------|-----|---------|------------|-----------------|------------|------|
| 1.0s | 0.38/0.48 | 0.45/0.59 | 0.41/0.52 | 0.48/0.61 | 0.45/0.53 | 0.49/0.66 | **0.30/0.40** |
| 2.0s | 0.63/0.93 | 0.76/1.06 | 0.67/0.94 | 0.76/1.08 | 0.72/0.96 | 0.77/1.11 | **0.58/0.88** |
| 3.0s | 0.94/1.34 | 1.06/1.40 | 0.98/1.35 | 1.06/1.43 | 1.00/1.39 | 1.11/1.46 | **0.87/1.31** |
| 4.0s | 1.17/1.61 | 1.32/1.74 | 1.18/1.63 | 1.35/1.78 | 1.19/1.66 | 1.37/1.79 | **1.13/1.60** |
| Acc. | 0.69 | 0.65 | 0.70 | 0.68 | 0.69 | 0.64 | **0.73** |

### 5.1.1 Datasets

**Stanford Drone Dataset.** This dataset consists of more than 11,000 persons in 20 scenes captured from the campus of Stanford University in bird's eye view. We follow the [27] standard train-test split, and predict the future 4.8s (12 frames) using past 3.2s (8 frames). Note that SDD dataset does not provide explicit pedestrian's action. Instead, we record them as an abstract encoding of the pedestrian's speed and location. The action contains [left, right, straight, turn around].

**NBA SportVU Dataset.** It is collected by NBA using the SportVU tracking system, which reports the trajectories of the ten players and the ball in real basketball games. Each trajectory contains the 2D positions and velocities of the offensive team, consisting of 5 players. We predict the future 10 timestamps (4.0s) based on the historical 5 timestamps (2.0s). Each player's action contains [left, right, straight, turn around, pass, shoot].

### 5.1.2 Metrics

Here, we adopt two metrics for evaluation: Average Displacement Error ($ADE$) and Final Displacement Error ($FDE$). Specifically, $ADE_N$ is defined as the minimum average distance between N predicted trajectories and the ground truth over all the involved entities within the prediction horizon. $FDE_N$ is defined as the minimum deviated distance of N predicted trajectories at the last predicted time step. Moreover, we also calculate the accuracy and F1 score of event-types predicted from each network.

## 5.2 Synthetic Experiment

We follow [10] to verify our model's rule discovery ability on synthetic datasets with a known set of ground-truth rules and weights. The synthetic events are generated from TLPPs [12] with a known set of rules and weights. We prepared 4 synthetic datasets, and each setting corresponds to different rule weights, rule length and number, type of temporal relation, and intensity of free predicates. Note that it was originally utilized for the temporal point process, so we modify it by adding spatial variables (such as "left, right, front, and behind") to fit in our settings. The weight learning results on 4 synthetic datasets are shown in the Figure 3, where 2400 sequences are used for evaluation. Each plot at the bottom compares the genuine rule weights to the learned rule weights and reports the Mean

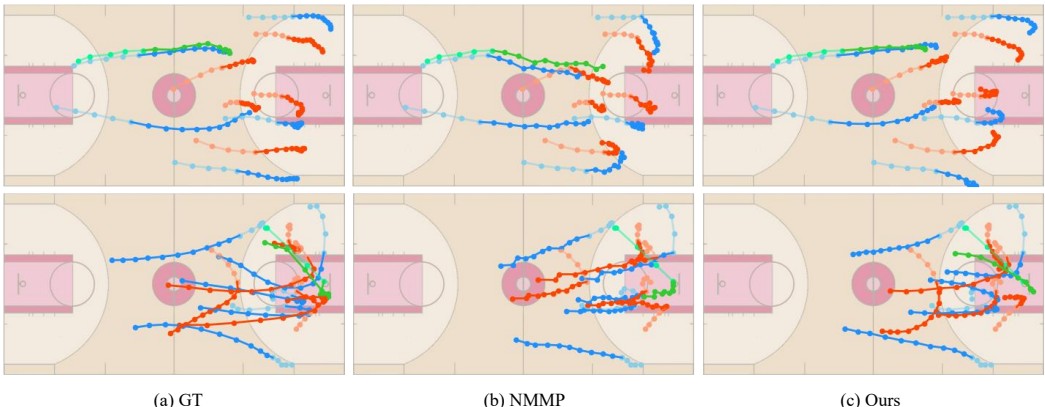

|              | (a) GT              | (b) NMMP            | (c) Ours            |

Figure 4: Visual results on the NBA dataset. We plot the predicted trajectories from the state-of-the-art method NMMP (b), Ours (c) and ground truth (a). The red/blue color represents players of two teams and the green color represents the basketball. Light color represents the past trajectory.

Table 2: Quantitative results ($ADE_{20}$, $FDE_{20}$ and F1 score) of trajectory prediction in SDD dataset. The bold/underlined font represent the best/second best result.

| Metrics | PECNet | Y-Net | MID | NMMP | STGAT | SOPHIE | Social SSL | NSP-SFM | STAR | Ours |
|---------|--------|-------|-----|------|-------|--------|------------|---------|------|------|
| $ADE_{20}$ | 20.03 | 7.85 | 7.61 | 14.12 | 14.43 | 15.56 | 6.63 | 6.52 | 10.76 | **6.41** |
| $FDE_{20}$ | 33.86 | 11.85 | 14.30 | 20.68 | 22.59 | 24.32 | 12.23 | 10.61 | 17.03 | **10.23** |
| F1 score | 0.37 | 0.54 | 0.49 | 0.41 | 0.33 | 0.39 | 0.53 | 0.58 | 0.56 | **0.59** |

Absolute Error (MAE). As we can see, almost all truth rule weights are learned correctly. It shows an accurate performance of our algorithm in terms of both the rule discovery and parameter learning.

## 5.3 Analysis

We compare our method with several state-of-the-art approaches, and Table 2 presents the qualitative results on the SDD dataset. The proposed model achieved the best performance in ADE, FDE and accuracy. We observe that our method significantly outperforms all baselines measured by ADE and FDE. It achieves an ADE of 6.41 and FDE of 10.23 at $K = 20$ in SDD datset, which exceeds the previous state-of-the-art performance of Y-Net [13] by 18.3% on ADE and 13.6% on FDE. In Table 1, our method also achieve higher performance than Y-Net in NBA dataset. This is because Y-Net firstly assume that the waypoint lies on a straight line segment connecting the sampled goal and the past trajectory, then use a multivariate Gaussian prior centered at the assumed location. This assumption can not well suit in other complex conditions, such as the trajectory of players in the NBA dataset.

Compared with MID [6], we also obtain 15.7% ADE and 28.4% FDE improvement. Note that they carefully design a Transformer-based architecture to model the temporal dependencies in trajectories, but ignore the spatial correlation of agents. Our transformer-based network aims at generating high-quality logic rules based on spatial-temporal relation under the principle to maximize the likelihood of the observational human actions. For NSP-SFM [30], it obtains high performance in SDD dataset but can not achieve the same level in NBA dataset. It combines physics with deep learning for trajectory prediction and accommodates arbitrary physics models. The limitation of it lies in specific physics models, such as pedestrians, and is deterministic. So it can not deal with some strategy-based conditions, including basketball and football games. But our logic-learning method tries to use a set of spatial-temporal logic rules with intention variables involved as principles to model the dynamics of human actions, not restrained into specific conditions.

Further, The proposed model achieved the best scores in F1 score, which is an balanced metric considering both recall and precision. This is because the rule generator and evaluator can collaborate with each other to reduce search space and learn better rules. More experiments and ablation studies can be found in Supplementary Material.

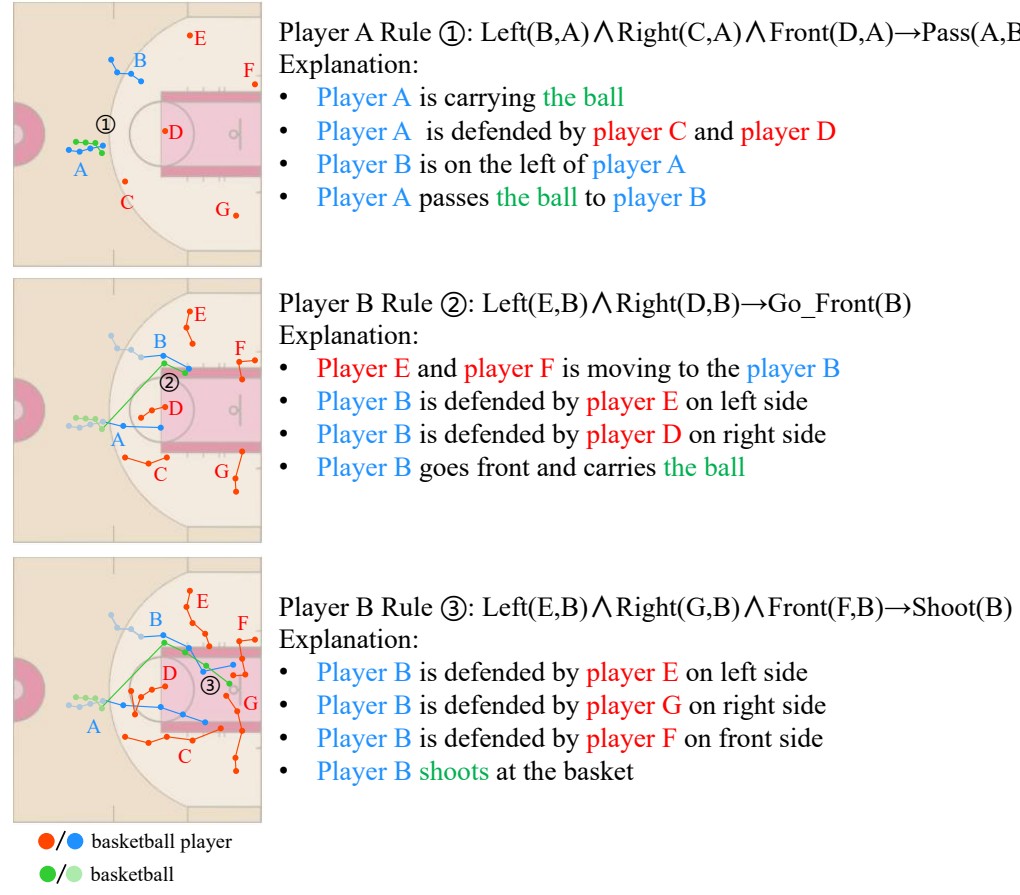

Player A Rule ①: Left(B,A)∧Right(C,A)∧Front(D,A)→Pass(A,B)
Explanation:
- Player A is carrying the ball
- Player A is defended by player C and player D
- Player B is on the left of player A
- Player A passes the ball to player B

Player B Rule ②: Left(E,B)∧Right(D,B)→Go_Front(B)
Explanation:
- Player E and player F is moving to the player B
- Player B is defended by player E on left side
- Player B is defended by player D on right side
- Player B goes front and carries the ball

Player B Rule ③: Left(E,B)∧Right(G,B)∧Front(F,B)→Shoot(B)
Explanation:
- Player B is defended by player E on left side
- Player B is defended by player G on right side
- Player B is defended by player F on front side
- Player B shoots at the basket

●/● basketball player
●/● basketball

Figure 5: Visualization and explanation of logic rules in NBA dataset.

To evaluate the robustness of our method, We added some noise and randomly removed several tracks in the NBA dataset, then evaluated all methods in Table 3 ("Ours*" is the original results without noises or missing tracks). As we can see, our method still achieves the best performance. Moreover, by comparing "Ours" and "Ours*", we can see that the quality of tracks has less influence on our method, which demonstrates the robustness of our method.

Our method not only outperforms all the above comparative methods in quantitative evaluations, but also produces visually pleasing results among them. Figure 4 compares the predicted trajectories of NMMP, ground-truth (GT) and our methods on NBA dataset. It is obvious that our method produces more precise predictions than state-of-the art method NMMP. This is because our proposed spatial-temporal logic rules can actually captures the spatial relation of the players with surrounding players, as well as the temporal ordering constraints of the events.

## 5.4 Generated Logic Rules

We add visualization and explanation about the logic rule and corresponding actions from NBA dataset in Figure 5. Note that the static spatial relation {Left(B,A)} represents that the player B is on the left of player A. And the dynamic spatial relation {Away(A,E)} means that player A is getting away from player E. We can see that these logic rules are meaningful and diverse. In this picture, player A is defended by two players and then passes the ball to player B. Player B goes front and crossover to bypass three defenders and shoot at the basket. Our rule can actually represent their offensive strategy. In fact, our framework can actually adapt to some complex motions, such as cutting toward the ball, because these spatial-temporal predicates are fed into the rule generator and evaluator to obtain high-quality rules to explain the intention of players.

Table 3: Robustness of our method in NBA dataset in some noise.

| Methods | 1.0 | | 2.0 | | 3.0 | | 4.0 | |
|---|---|---|---|---|---|---|---|---|
| | ADE | FDE | ADE | FDE | ADE | FDE | ADE | FDE |
| Y-Net | 0.51 | 0.62 | 0.81 | 1.11 | 1.08 | 1.49 | 1.37 | 1.86 |
| NSP-SFM | 0.48 | 0.63 | 0.82 | 1.13 | 1.06 | 1.46 | 1.36 | 1.82 |
| Ours | 0.31 | 0.48 | 0.67 | 0.91 | 0.93 | 1.39 | 1.18 | 1.66 |
| Ours* | 0.30 | 0.40 | 0.58 | 0.88 | 0.87 | 1.31 | 1.13 | 1.60 |

## 6 Limitation

It's challenging to define some complex predicates with richer meanings for different datasets. Given a more informative dataset, our method can discover more principles-like complex rules. Our motivation is to consider the spatial-temporal relation between pedestrians and generate some high-quality logic rules to explain their behaviors. Although we only choose some simple actions in our experiments, they can bring some benefits for understanding the principles of biological agents' behaviors. Our framework is suitable for more complex conditions, supposing more sophisticated action predicates can be obtained from the data.

## 7 Conclusion

We proposed a framework for learning intrinsic spatial-temporal logic rules for explaining human actions. We regard logic rules as latent variables, and the rule generator as well as the rule evaluator are jointly learned with EM-based algorithm. In the experiments, our method can analyze the biological movement sequence of pedestrians and players, and obtained novel insights from generated logic rules. In the future, we plan to incorporate other physical laws into the models, such as conservation of energy and momentum to enhance robustness of our model.

## 8 Acknowledgement

Shuang Li's research was in part supported by the NSFC under grant No. 62206236, Shenzhen Science and Technology Program JCYJ20210324120011032, National Key R&D Program of China under grant No. 2022ZD0116004, and Guangdong Key Lab of Mathematical Foundations for Artificial Intelligence.

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
