# Discovering Intrinsic Spatial-Temporal Logic Rules to Explain Human Actions
# Supplementary Material

**Chengzhi Cao**[1,2], **Chao Yang**[2], **Ruimao Zhang**[2], **Shuang Li**[2*]
[1]University of Science and Technology of China,
[2]The Chinese University of Hong Kong (Shenzhen)
chengzhicao@mail.ustc.edu.cn, 222043011@link.cuhk.edu.cn,
zhangruimao@cuhk.edu.cn, lishuang@cuhk.edu.cn

## 1 Model Definition

**Static spatial relation predicates.** We define the spatial relations of two objects, such as {left, right, in front, behind, far from, inside}. Take "left" for example,

$$R_{\text{left}}(s_1, s_2) = \mathbb{1}\left\{\epsilon < \|s_1 - s_2\| < L, \operatorname{atan2}(s_1 - s_2) \in (\frac{3\pi}{4}, \pi] \cup [-\pi, -\frac{3\pi}{4})\right\}.$$

The other relations can also be represented in the same way. We will either treat the static spatial relation predicates as a boolean variable, or we can parameterize them as spatial kernel functions of $s_1, s_2$ with learnable parameters that map to $[0, 1]$.

**Dynamic spatial relation predicates.** We define dynamic spatial relations of two objects, such as {closer to, father away}. For example,

$$R_{\text{CloserTo}}((c_1, t, s_1), (c_2, t, s_2)) = \mathbb{1}\left\{\frac{\partial d}{dt} < 0\right\},$$

$$R_{\text{FartherAway}}((c_1, t, s_1), (c_2, t, s_2)) = \mathbb{1}\left\{\frac{\partial d}{dt} > 0\right\},$$

where $d = \|s_1 - s_2\|$.

One can freely define other types of spatial-temporal predicates. We just provide some concrete examples above to illustrate the ideas.

## 2 The Number of Training Triplets

To better evaluate different methods under cases where training triplets are limited, in this section we reduce the amount of training data to see how the performance varies. The results are presented in Figure 1. We can see that all of methods have better performance with the increase of training triplets. And our model achieves the best results.

## 3 Comparison of Parameters

The parameters and FLOPs of all methods is shown in Table 1. As we can see, with reasonable storage consumption, our method has comparable FLOPs and provides promising performance.

---

[*]Corresponding author.

37th Conference on Neural Information Processing Systems (NeurIPS 2023).

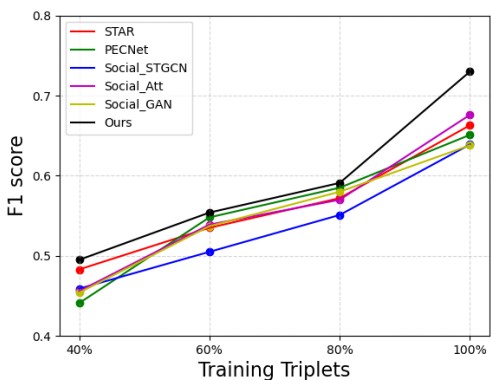

Figure 1: Performance w.r.t. training triplets.

Table 1: Comparison of different backbones on parameter numbers and FLOPS.

| Methods | # Params | FLOPS |
|---|---|---|
| PECNet | 9.38M | 764.03M |
| Social SSL | 2.27M | 24.36M |
| NMMP | 58.01M | 712.64M |
| STGAT | 1.10M | 34.32M |
| SOPHIE | 0.29M | 108.57M |
| STAR | 3.91M | 272.82M |
| Ours | 1.41M | 100.53M |

## 4  Backbone

We added the relevant ablation experiments on the different components of the approach. For the rule generator and the decoder, we compare ours (transformer-based) with three widely used backbones, including CNN, RNN and GNN (graph neural network), and evaluate them in the NBA dataset. As shown in Table 3 and Table 4, our architecture can actually achieve superior results in all metrics. Moreover, in the E-step, we will identify top K rules from all generated rules, where K is a tunable hyperparameter. So we also add a hyperparameter tuning study of the number of K. The results are shown in Table 2. The best result appears when K is set as 5, and the performance is almost the same after the K is larger than 5, but it brings more storage consumption. So finally, we set K as 5.

## 5  Additional Results

To make our current experimental results more convincing, we further added seven more recent SOTA baselines, including Y-Net [10], MID [5], NSP-SFM [15], Social-SSL [13], Social-Implicit [11], Social-VAE [14], and ABC+ [7]. Note that all of these newly included baselines were proposed in between 2021 and 2022. We also evaluated them in the ETH/UCY dataset, as shown in Table 5 respectively. These new experimental results show that we still achieve superior results in most metrics.

Table 2: Comparison of different values of K.

| K | 1.0 | | 2.0 | | 3.0 | | 4.0 | |
|---|---|---|---|---|---|---|---|---|
| | ADE | FDE | ADE | FDE | ADE | FDE | ADE | FDE |
| 1 | 0.47 | 0.53 | 0.67 | 1.08 | 1.00 | 1.41 | 1.27 | 1.71 |
| 2 | 0.42 | 0.47 | 0.66 | 1.04 | 1.00 | 1.40 | 1.23 | 1.68 |
| 3 | 0.40 | 0.45 | 0.61 | 0.99 | 0.96 | 1.35 | 1.21 | 1.65 |
| 4 | 0.34 | 0.40 | 0.59 | 0.94 | 0.92 | 1.32 | 1.19 | 1.63 |
| 5 | **0.30** | **0.40** | **0.58** | **0.88** | **0.87** | **1.31** | **1.13** | **1.60** |
| 6 | 0.30 | 0.41 | 0.59 | 0.89 | 0.87 | 1.32 | 1.13 | 1.61 |

Table 3: Comparison of different backbones in the rule generator in SDD dataset. The bold font represent the best result.

| Method | 1.0 | | 2.0 | | 3.0 | | 4.0 | |
|--------|-----|-----|-----|-----|-----|-----|-----|-----|
| | ADE | FDE | ADE | FDE | ADE | FDE | ADE | FDE |
| CNN | 0.41 | 0.60 | 0.78 | 1.07 | 0.98 | 1.53 | 1.24 | 1.76 |
| GNN | 0.38 | 0.59 | 0.76 | 1.03 | 0.96 | 1.49 | 1.19 | 1.69 |
| RNN | 0.36 | 0.49 | 0.69 | 1.00 | 0.95 | 1.39 | 1.17 | 1.67 |
| Ours | **0.30** | **0.40** | **0.58** | **0.88** | **0.87** | **1.31** | **1.13** | **1.60** |

Table 4: Comparison of different backbones in the decoder in SDD dataset. The bold font represent the best result.

| Method | 1.0 | | 2.0 | | 3.0 | | 4.0 | |
|--------|-----|-----|-----|-----|-----|-----|-----|-----|
| | ADE | FDE | ADE | FDE | ADE | FDE | ADE | FDE |
| CNN | 0.39 | 0.47 | 0.74 | 1.01 | 0.93 | 1.49 | 1.23 | 1.80 |
| GNN | 0.36 | 0.41 | 0.72 | 0.96 | 0.92 | 1.42 | 1.19 | 1.72 |
| RNN | 0.35 | 0.40 | 0.66 | 0.93 | 0.91 | 1.34 | 1.18 | 1.65 |
| Ours | **0.30** | **0.40** | **0.58** | **0.88** | **0.87** | **1.31** | **1.13** | **1.60** |

# 6 Implement Details

The input of our framework is the historical trajectory coordinates of entities, and the output is the predicted future trajectory. Coordinates as input would be first encoded into a vector by three-layer MLPs before being fed into the Transformer, and a ReLU nonlinearity following each of the first two layers. The dimensions of keys, values, and queries are all set to 256, and the hidden dimension of feed-forward layers is 512. The number of heads for multi-head attention is 8.

# 7 Automatically Generating Actions

To learn specific actions would require learning recursion and predicate invention. Invented predicates can be interpreted as a set of phrases to express the meaning of actions. we embed meta-interpretive learning (MIL), which supports efficient predicate invention and learning of recursive logic programs built as a set of metalogical substitutions by a modified Prolog meta-interpreter, into our framework and evaluate its performance in the NBA dataset. Obviously, this operation actually brings slight improvements.

# 8 Determine the Number of Rules

The number of rules can also be automatically learned by neural networks or other deep learning methods. In our settings, we manually set K as the upper limit to reduce the computation cost. In E-step, some movements are so simple that they only require less rules because most of the candidate rules' weights are minimal. Moreover, we set a weight threshold to automatically determine the

Table 5: Quantitative results ($ADE_{20}/FDE_{20}$) of trajectory prediction in ETH/UCY dataset. The bold/underlined font represent the best/second best result.

| Methods | ETH | HOTEL | UNIV | ZARA1 | ZARA2 | AVG |
|---------|-----|-------|------|-------|-------|-----|
| Y-Net | 0.28/0.33 | 0.10/0.14 | 0.24/0.41 | 0.17/0.27 | 0.13/0.22 | 0.18/0.27 |
| MID | 0.39/0.66 | 0.13/0.22 | 0.22/0.45 | 0.17/0.30 | 0.13/0.27 | 0.21/0.38 |
| NSP-SFM | 0.25/0.44 | 0.09/0.13 | 0.21/0.38 | 0.16/0.27 | 0.12/0.20 | 0.17/0.24 |
| Social SSL | 0.69/1.37 | 0.24/0.44 | 0.51/0.93 | 0.42/0.84 | 0.34/0.67 | 0.44/0.85 |
| Social Implicit | 0.66/1.44 | 0.20/0.36 | 0.32/0.60 | 0.25/0.50 | 0.22/0.43 | 0.33/0.37 |
| Social-VAE | 0.41/0.58 | 0.13/0.19 | 0.21/0.36 | 0.17/0.29 | 0.13/0.22 | 0.21/0.33 |
| ABC+ | 0.31/0.44 | 0.16/0.21 | 0.25/0.47 | 0.21/0.28 | 0.20/0.26 | 0.23/0.32 |
| Ours | **0.22/0.30** | **0.07/0.13** | **0.16/0.34** | **0.14/0.25** | **0.07/0.16** | **0.13/0.24** |

Table 6: Quantitative results (ADE/FDE) of trajectory prediction in NBA dataset. "Ours+" means that we automatically extract the actions.

| Times | Ours | Ours+ |
|---|---|---|
| 1.0s | 0.30/0.40 | 0.28/0.37 |
| 2.0s | 0.58/0.88 | 0.58/0.87 |
| 3.0s | 0.87/1.31 | 0.85/1.28 |
| 4.0s | 1.13/1.60 | 1.11/1.54 |

Table 7: Ablation study of discretization rate in SDD dataset.

| Metric | d=0.1 | d=0.2 | d=0.3 | d=0.4 |
|---|---|---|---|---|
| ADE | 8.24 | 12.21 | 15.03 | 19.49 |
| FDE | 1.74 | 19.58 | 24.98 | 30.87 |

number of rules, and the results are shown in the Table 8. Obviously, this operation can bring slight improvements.

## 9  Discretization Rate

The discretization rate can directly influence the computation cost and the loss of information. The larger discretization rate means more loss of information but less computation cost. In fact, our method can tackle different discretization rates properly. We also set different discretization rates (presented as d, and "d=0.1" means that we remove 10% trajectory data regularly) in the SDD dataset and obtain some results in Table 7.

## 10  3D scenarios

To demonstrate the benefit of our method in 3D scenarios, we also evaluate it in the 3D motion prediction dataset, called Haggling dataset, over 7 different action types, and the results are shown in Table 9. Our method achieves promising results for both short-term and long-term predictions of complex activities.

## 11  Related work

**Trajectory Prediction**   A number of recent works [1, 3, 8] have studied to incorporate the interaction relations of multiple objects to infer trajectories. Social-LSTM [1] is a pioneering work that models social interaction for trajectory prediction via the concept of social pooling, which aggregates the hidden states among the nearby agents. SocialGAN [6] revises the pooling module to capture global information from all agents in the scene. Different from previous works that utilize LSTM models, a recent line of studies introduced new ideas, such as graph attention networks [10, 14], for modeling the social interactions between pedestrians. Unlike previous works, we introduce a novel approach on the spatio-temporal modeling strategy under guidance of logic rules. This makes better use of the sequence data by predicting the agent's hidden intentions and modeling the spatio-temporal

Table 8: Quantitative results of trajectory prediction in NBA dataset. "Ours++" means that we automatically determine the number of rules.

| Times | Ours | Ours++ |
|---|---|---|
| 1.0s | 0.30/0.40 | 0.29/0.38 |
| 2.0s | 0.58/0.88 | 0.57/0.86 |
| 3.0s | 0.87/1.31 | 0.85/1.29 |
| 4.0s | 1.13/1.60 | 1.11/1.55 |

Table 9: Performance evaluation (MAE) of comparison methods on the Haggling (H3.6m) dataset.

| Time | 80ms | 160ms | 320ms | 400ms | 560ms | 640ms | 720ms | 1000ms |
|---|---|---|---|---|---|---|---|---|
| Walking | 0.38 | 0.58 | 0.80 | 0.89 | 0.98 | 1.03 | 1.11 | 1.22 |
| Eating | 0.25 | 0.39 | 0.60 | 0.76 | 0.94 | 1.01 | 1.03 | 1.29 |
| Discussing | 0.31 | 0.57 | 0.88 | 0.99 | 1.48 | 1.65 | 1.81 | 1.96 |
| Phoning | 0.55 | 0.83 | 1.22 | 1.35 | 1.58 | 1.65 | 1.72 | 1.92 |
| Posing | 0.27 | 0.56 | 1.19 | 1.48 | 1.93 | 2.14 | 2.29 | 2.58 |
| Sitting | 0.40 | 0.63 | 1.02 | 1.18 | 1.28 | 1.34 | 1.40 | 2.02 |
| Waiting | 0.36 | 0.69 | 1.25 | 1.46 | 1.80 | 1.95 | 2.12 | 2.57 |

interactions between agents. Casas et al. [2] introduced IntentNet, a novel deep network that reasons about both high level behavior and long term trajectories, which exploited motion and prior knowledge about the road topology. Li et al. [9] proposed a generic trajectory forecasting framework with explicit interaction modeling via a latent graph, and evolve the underlying interaction graph adaptively along time. Graber et al. [4] developed Dynamic Neural Relational Inference by incorporating insights from sequential latent variable models to predict separate relation graphs for every timestamp. Tang et al. [12] introduced a collaborative-uncertainty-based framework to models the uncertainty from the usage of interaction modules in multi-agent trajectory forecasting.