# OpenReview forum: "Discovering Intrinsic Spatial-Temporal Logic Rules to Explain Human Actions"
_NeurIPS.cc/2023/Conference — NeurIPS 2023 poster_

### Official Review · Reviewer_AEbP · 2023-06-24

**Soundness:** 3 good
**Presentation:** 2 fair
**Contribution:** 3 good
**Rating:** 7
**Confidence:** 3

**Summary:**

This paper proposes a model for inferring rules based on observation trajectories of (entity, time, location), with the aim of maximizing the probabilities of certain "events". The aim is to predict the next events based on past trajectories, conditioned on the latent "rules" which need to be marginalized over. But this is intractable, so an "encoder" or "recognition" network is used, similar to the logic of variational autoencoder. In the encoder, a transformer is used to define the probability distribution over possible rules based on input trajectories, and a decoder can sample these rules to generate the next event. The model is used for trajectory prediction tasks on two datasets with better results than existing trajectory prediction model. Examples of rules learnt are also given in case of one dataset.

**Strengths:**

1) A nice framework is proposed for representation of spatio-temporal action rules based on logic
2) The "rules" are expressed as a latent variable model, where inference can be done using E-M algorithm applied to an encoder-decoder model
3) Strong results are shown on trajectory prediction tasks, sample rules are also shown

In my opinion, the rule definition and rule inference framework is quite general and can find wider applications than the experiments described in the paper.

**Weaknesses:**

It is not very clear how exactly the goal states are estimated (eg. in Fig 3), or how the future trajectories are generated (Fig 3 and experiments), I get the broad idea, but precision is missing.

**Questions:**

1) Can we have an experiment on synthetic setting with ground truth data generated according to specific rules (eg. like a cellular automata), and then show if the proposed model can recover the rules?
2) Please add an algorithm to explain how the trajectory prediction is carried out, step-by-step.
3) What can be other applications of this framework beyond trajectory prediction? Also, can the "rules" here be related to "policies" to choose actions as in Reinforcement Learning?
4) Can the rule probability distribution be modelled by a cheaper model than Transformer?

---

> ### Author Rebuttal · Authors · 2023-08-08
>
> We sincerely thank you for your recognition of our work and your insightful reviews! We hope our responses can address your questions. Our responses are listed below.
>
> **Q1: It is not very clear how exactly the goal states are estimated (eg. in Fig 3), or how the future trajectories are generated.**
>
> A1: Following [1][2], the goal and waypoint heatmap are extracted from the decoder in our framework. We estimate distributions of future waypoint positions which along with the goal points are used to obtain explicit maps over all the remaining intermediate trajectory positions.
>
> [1] Mangalam K, An Y, Girase H, et al. From goals, waypoints \& paths to long-term human trajectory forecasting[C]//Proceedings of the IEEE/CVF International Conference on Computer Vision. 2021: 15233-15242.
>
> [2] Jacobs H O, Hughes O K, Johnson-Roberson M, et al. Real-time certified probabilistic pedestrian forecasting[J]. IEEE Robotics and Automation Letters, 2017, 2(4): 2064-2071.
>
> **Q2: Can we have an experiment on the synthetic setting with ground truth data generated according to specific rules (eg. cellular automata), and then show if the proposed model can recover the rules?**
>
> A2: Yes. Thanks for your advice. We follow [3] to verify our model’s rule discovery ability on synthetic datasets with a known set of ground-truth rules and weights. Note that it was originally utilized for the temporal point process, so we modify it by adding spatial variables (such as “left, right, front, and behind”) to fit in our settings. The weight learning results on 4 synthetic datasets are shown in the following table. More results are  shown **in the attached PDF Figure 2.**
>
>
> | Weight  | Dataset-1 | Dataset-2 | Dataset-3 |Dataset-4|
> |--|--|--|--|--|
> | w0 | 1.0/0.98 | 0.5/0.45 |1.5/1.47|2.0/1.82|
> | w1 | 1.0/0.91 | 0.5/0.40 |1.5/1.44|1.0/0.97|
> | w2 | 1.0/0.81 | 0.5/0.34 |1.5/1.39|1.0/0.92|
>
> Table: Rule discovery and weight learning results (GT weights/learned weights) on 4 synthetic datasets.
>
> [3] Li S, Feng M, Wang L, et al. Explaining point processes by learning interpretable temporal logic rules[C]//International Conference on Learning Representations. 2021.
>
> **Q3: Please add an algorithm to explain how the trajectory prediction is carried out, step-by-step.**
>
> A3: Following the reviewer's advice, we explain the trajectory prediction step by step.
>
> ---
>
> **Input**: observed history trajectory $H$ of entity $v$, time horizon $T$, the number of rules $N$
>
> **while** not converge **do**
>
> $\qquad$ Use the rule generator $p_\theta$ to generate a set of rules $z$ by $p_\theta(z | v, H_t) = Ψ(z|N, Trans_\theta(v, H_t))$;
>
> $\qquad$    Update the predictor $p_w$ based on generated rules $z$;
>
> $\qquad$    Identify K high-quality rules from $z$ according to Eq.10;
>
> $\qquad$    Update the rule generator $p_\theta$ according to the identified rules;
>
> **end**
>
>  Use $p_\theta$ to generate rules and feed them into $p_w$ for prediction;
>
> **for** $i = 1,\dots,T$ **do**
>
> $\qquad$    Calculate posterior probability $P(z_i|CurrentPosition)$;
>
> **end**
>
> Update parameters based on computed posterior probabilities;
>
> **for $i = 1,\dots,T$ do**
>
> $\qquad$    Compute the weighted average of predicted positions from each component;
>
> **end**
>
> ---
>
> **Q4: What can be other applications of this framework beyond trajectory prediction? Also, can the "rules" here be related to "policies" to choose actions as in Reinforcement Learning?**
>
> A4: (1) This framework can also be applied to other spatial-temporal events, such as social analysis, mobile robots and epidemic forecasting. (2) Yes, the generated rule can be utilized to choose actions in Reinforcement Learning. It's our future work to represent the policies in reinforcement learning by first-order logic based on policy gradient methods and differentiable inductive logic programming, which has significant advantages in terms of interpretability and generalisability.
>
>
> **Q5: Can the rule probability distribution be modeled by a cheaper model than Transformer?**
>
> A5: Yes, the rule probability distribution be modeled by other backbones, such as CNN. In the supplementary material Section 7, we have compared our method (transformer-based) with three widely used backbones, including CNN, RNN, and GNN (graph neural network), and evaluated them in the NBA dataset. The results have been shown **in supplementary material Table 3**. Obviously, our architecture can actually achieve superior results in all metrics. Below we show the results for the reviewer’s convenience.
>
> | Method  | 1.0s | 2.0s | 3.0s | 4.0s|
> |--|--|--|--|--|
> | CNN | 0.41/0.60 | 0.78/1.07 |0.98/1.53|1.24/1.76|
> | GNN | 0.38/0.59 | 0.76/1.03 |0.96/1.49|1.19/1.69|
> | RNN | 0.36/0.49 | 0.69/1.00 |0.95/1.39|1.17/1.67|
> | Ours |0.30/0.40 | 0.58/0.88 |0.87/1.31|1.13/1.60|
>
> Table: Comparison of different backbones in the rule generator in the SDD dataset.

---

> > ### Comment · Reviewer_AEbP · 2023-08-11
> >
> > I thank the authors for the responses. I am quite satisfied and have no further questions. I will be happy if these are added to the final version of the paper if accepted.

---

### Official Review · Reviewer_7WeJ · 2023-07-05

**Soundness:** 4 excellent
**Presentation:** 4 excellent
**Contribution:** 3 good
**Rating:** 7
**Confidence:** 3

**Summary:**

This paper presents a novel approach for human trajectory prediction, introducing a learnable rule-based framework that combines rule generation/reasoning and EM optimization. Unlike previous works in the field, this framework utilizes a neural rule generator to generate rules and treats them as latent variables. Additionally, it selects the top k rules from the entire rule set to predict future trajectories. The experimental results on two datasets demonstrate the effectiveness of the proposed framework.

**Strengths:**

- The paper is well-structured and written in a clear manner, making it easy to follow. The authors have used appropriate notation and included equations where necessary, enhancing the readability of the paper.

- The detailed analysis and visualizations presented in sections 5.4-5.6 and the ones in the supplementary materials contribute to a better understanding of the proposed method. I found them enjoyable and informative to read.

- The paper introduces a novel framework that incorporates separate steps for rule generation and logic reasoning. This approach is original and reasonable for the given task.

- The proposed framework demonstrates improved results on the Stanford Drone Dataset and NBA SportVU dataset.

- The authors have put effort into providing numerous illustrative examples that showcase the behavior of the proposed framework. The explanation of logic rules in the NBA dataset (Figure 5) is particularly interesting.

**Weaknesses:**

- The predicates are required to be manually defined and these predicates may need to be redefined when applied to new scenarios, such as transitioning from 2D to 3D scenarios.

- ETH/UCY benchmarks have been widely used for benchmarking short-term trajectory prediction but missing from this paper.

**Questions:**

- Regarding the latent rule space, what is the dimension of the latent space z?

- I have reservations about fully accepting the argument presented in L207 that "our method can discover more principles-like complex rules." The generation of more rules for complex conditions necessitates a larger latent space (higher dimension), which can pose challenges in generating high-quality rules.

- In L207-L208, it is mentioned that "For each query, we aim to identify top K rules $z_I$ from all generated rules $\hat{z}$," with K set to 5 in the experiments. Not all movements require up to K rules. Some movements may be simple and only require 1-2 rules (as seen in the second example in Figure 2). Is it possible to automatically determine the number of rules based on their weights?

**Limitations:**

The authors have provided a discussion of the limitations in Section 6.

---

> ### Author Rebuttal · Authors · 2023-08-08
>
> We sincerely thank you for your recognition of our work and your insightful reviews! We hope our responses can address your questions. Our responses are listed below.
>
> **Q1: The predicates are required to be manually defined and these predicates may need to be redefined when applied to new scenarios, such as transitioning from 2D to 3D scenarios.**
>
> A1: Thanks for your advice. To demonstrate the benefit of our method in 3D scenarios, we also evaluate it in the 3D motion prediction dataset, called Haggling dataset [1], over 7 different action types, and the results are shown in Table 1. Our method achieves promising results for both short-term and long-term predictions of complex activities.
>
> |Time|80ms|160ms|320ms|400ms|560ms|640ms|720ms|1000ms|
> |--|--|--|--|--|--|--|--|--|
> |Walking|0.38|0.58|0.80|0.89|0.98|1.03|1.11|1.22|
> |Eating|0.25|0.39|0.60|0.76|0.94|1.01|1.03|1.29|
> |Discussion|0.31|0.57|0.88|0.99|1.48|1.65|1.81|1.96|
> |Phoning|0.55|0.83|1.22|1.35|1.58|1.65|1.72|1.92|
> |Posing|0.27|0.56|1.19|1.48|1.93|2.14|2.29|2.58|
> |Sitting|0.40|0.63|1.02|1.18|1.28|1.34|1.40|2.02|
> |Waiting|0.36|0.69|1.25|1.46|1.80|1.95|2.12|2.57|
>
> Table: Performance evaluation (in MAE) of comparison methods over on the Haggling (H3.6) dataset.
>
> [1] Liu Z, Wu S, Jin S, et al. Towards natural and accurate future motion prediction of humans and animals[C]//Proceedings of the IEEE/CVF Conference on Computer Vision and Pattern Recognition. 2019: 10004-10012.
>
> **Q2: ETH/UCY benchmarks have been widely used for benchmarking short-term trajectory prediction but missing from this paper.**
>
> A2: We have evaluated our method in the ETH/UCY dataset, and **due to the page limitation, we put the result in the supplementary material Table 6**. Below we show the results for the reviewer’s convenience. These new experimental results show that we still achieve superior results in most metrics.
>
> | Methods  | ETH | HOTEL | UNIV | ZARA1 | ZARA2 |AVG|
> |--|--|--|--|--|--|--|
> |Y-Net          |0.28/0.33|0.10/0.14|0.24/0.41|0.17/0.27|0.13/0.22|0.18/0.27|
> |MID            |0.39/0.66|0.13/0.22|0.22/0.45|0.17/0.30|0.13/0.27|0.21/0.38|
> |NSP-SFM        |0.25/0.44|0.09/0.13|0.21/0.38|0.16/0.27|0.12/0.20|0.17/0.24|
> |Social SSL     |0.69/1.37|0.24/0.44|0.51/0.93|0.42/0.84|0.34/0.67|0.44/0.85|
> |Social Implicit|0.66/1.44|0.20/0.36|0.32/0.60|0.25/0.50|0.22/0.43|0.33/0.37|
> |Social-VAE     |0.41/0.58|0.13/0.19|0.21/0.36|0.17/0.29|0.13/0.22|0.21/0.33|
> |ABC+           |0.31/0.44|0.16/0.21|0.25/0.47|0.21/0.28|0.20/0.26|0.23/0.32|
> |Ours           |**0.22/0.30**|**0.07/0.13**|**0.16/0.34**|**0.14/0.25**|**0.07/0.16**|**0.13/0.24**|
>
> Table: Quantitative results (ADE20/F DE20) of trajectory prediction in ETH/UCY dataset. The
> bold font represents the best result.
>
> **Q3: Regarding the latent rule space, what is the dimension of the latent space z?**
>
> A3: The dimension of latent embedding in our method is set to 32 on all datasets. We will add it in a revision.
>
> **Q4:I have reservations about fully accepting the argument presented in L207.**
>
> A4: Thanks for your advice. Complex rule generation actually requires higher-dimensional latent space, which is also our future work. We will modify this sentence as “our method has the potential to discover more principles-like complex rules”.
>
> **Q5: Is it possible to automatically determine the number of rules based on their weights?**
>
> A5: Yes, the number of rules can also be automatically learned by neural networks or other deep learning methods. In our settings, we manually set K as the upper limit to reduce the computation cost. In E-step, some movements are so simple that they only require less rules because most of the candidate rules’ weights are minimal. Moreover, following the reviewer’s advice, we set a weight threshold to automatically determine the number of rules, and the results are shown in the following table. Obviously, this operation can bring slight improvements.
>
>
> | Times  | Ours | Ours++ |
> |--|--|--|
> | 1.0s | 0.30/0.40 | 0.29/0.38 |
> | 2.0s | 0.58/0.88 | 0.57/0.86 |
> | 3.0s | 0.87/1.31 | 0.85/1.29 |
> | 4.0s | 1.13/1.60 | 1.11/1.55 |
>
> Table: Quantitative results of trajectory prediction in NBA dataset. “Ours++” means that we automatically determine the number of rules.

---

> > ### Comment · Reviewer_7WeJ · 2023-08-15
> >
> > Thank you for your response. The additional details you provided have effectively addressed my concerns. Their inclusion will undoubtedly enhance the quality of the paper. I'm therefore pleased to recommend acceptance.

---

### Official Review · Reviewer_zUPP · 2023-07-06

**Soundness:** 2 fair
**Presentation:** 3 good
**Contribution:** 3 good
**Rating:** 6
**Confidence:** 2

**Summary:**

The paper proses a method for learning spatio-temporal logic rules to explain human actions, utilizing the EM framework. The method results are more easy interpretable by humans, thanks to the logic rules. The method beats some state-of-the-art methods on two real-world motion prediction datasets.

**Strengths:**

The method is well-motived, i.e. that human actions are driven by intention as well as social and environmental factors. The explainability of the model is impressive and well-visualized.

**Weaknesses:**

My major concern with this work is the need for dataset-depended actions. Biasing the model with additional labels, which are not provided to other methods, allows it to spend more capacity on forecasting, effectively producing an unfair advantage. It would have been better if the authors provide an approach that can automatically extract the actions, without human pre-selection.

While the general intention is well-motivated, the method overview and method motivation in the introduction section is not well described, making it difficult to get a first good understanding of what the method tries to achieve and how this is done. For example, the authors explain that “[] logic rules present a compact knowledge representation” (L19) but they do not further explain what “logic” they are using and how it is defined.

In the method section a lot of important details are missing, making it difficult to follow:
* In the Equations in L104.5 and L113.5 the authors utilize variable v and v’ without introducing what this variable means. The authors should explain the variable appropriately.
* In L122 the authors introduce various new object types “person”, “block” and “key” which are not explained.
* In L117 the paper says that the binary predicated values can be softened into “probabilistic” ones using kernel functions without details or reference to other parts of the paper. This makes it particularly difficult to understand as the paper has not yet revealed how the logic rules are applied to obtain the predictions.

Spatio-temporal predicates (L104) contain the time and space dimension. This necessitates that both time and area must be discretized. The discretization rate would be an important hyper-parameter and should be ablated accordingly.

Why do the authors not follow the experimental setup in [23] and also evaluate on the ETH-UCY dataset? Can the dataset not be labeled with actions? If so: why? What makes the dataset difficult for this method setup?

The organization of the experiments section is not convincing: while the authors conducted a lot of experiments they should have added them into the main paper.

**Minor issues**:

The authors claim their models “superior interpretability” (L15) but do not properly explained in text (only attempted in Figure 5). This however should be explained briefly.

Equations should be numbered to make referencing them easier

**Questions:**

It seems that using the full 3D human pose would be a much better indicator for human actions, interactions with objects, and interactions with other persons. Why did the authors use only trajectory data? For example, the Haggling dataset [a] contains triadic interactions under a well-defined social protocol which could have been utilized.

L94: “Consider a set of objects denoted as C”: does object mean humans? This should be explicitly stated if true.

L96: could the authors elaborate how k is encoded?

—

[a] Liu, Zhenguang, et al. "Towards natural and accurate future motion prediction of humans and animals.” CVPR 2019.

**Limitations:**

The methods challenges are discussed to some degree.

---

> ### Author Rebuttal · Authors · 2023-08-08
>
> We sincerely thank you for your insightful reviews! We hope our response below addresses your concerns.
>
> **Q1: My major concern with this work is the need for dataset-depended actions**
>
> A1: Thanks for your suggestion. To learn specific actions would require learning recursion and predicate invention [a]. Invented predicates can be interpreted as a set of phrases to express the meaning of actions. Following [a], we embed meta-interpretive learning (MIL), which supports efficient predicate invention and learning of recursive logic programs built as a set of metalogical substitutions by a modified Prolog meta-interpreter, into our framework and evaluate its performance in the NBA dataset. Obviously, this operation actually brings slight improvements.
>
>
> | Times  | Ours | Ours+ |
> |--|--|--|
> | 1.0s | 0.30/0.40 | 0.28/0.37 |
> | 2.0s | 0.58/0.88 | 0.58/0.87 |
> | 3.0s | 0.87/1.31 | 0.85/1.28 |
> | 4.0s | 1.13/1.60 | 1.11/1.54 |
>
> Table: Quantitative results (ADE/FDE) of trajectory prediction in NBA dataset. “Ours+” means that we automatically extract the actions.
>
> [a] Muggleton S H. Meta-interpretive learning of higher-order dyadic datalog: Predicate invention revisited[J]. Machine Learning, 2015.
>
>
> **Q2: They do not further explain what “logic” they are using and how it is defined.**
>
> A2: Indeed, “logic” emphasizes high-level reasoning, and encourages structuring the world in terms of objects, properties, and relations [b]. “logic” can provide the formal machinery to reason about some concepts, such as time, space, abstraction, and causality, in a rigorous way.
>
> [b] Belle V. Symbolic logic meets machine learning: A brief survey in infinite domains[C]//International conference on scalable uncertainty management. 2020.
>
> **Q3: In the method section a lot of details are missing ...**
>
> A3: (1) v and v’ represent the entity-time-location triplet (described in Line 125), so X(v) and R(v,v’) are logic random variables that define the property or relation of entities. (2) “person”, “block” and “key” are the specific instances in the object set C, so that in Line 121, this rule represents that a person wants to pick up a key, while one block is in front of him and the key is behind him, so he turns around. We utilize this example to manifest the meaning of logic rules. (3) Following [c], we use the softened representation and a weighted combination of logic rules to deal with uncertainty in events. The soft constraint has been utilized in other papers (Ref [10] [24] in the main paper).
>
> [c] Li S, et al. Temporal logic point processes[C]// ICML, 2020.
>
> **Q4: The discretization rate would be an important hyper-parameter and should be ablated accordingly.**
>
> A4: The discretization rate can directly influence the computation cost and the loss of information. The larger discretization rate means more loss of information but less computation cost. In fact, our method can tackle different discretization rates properly. In the main paper, we set the smallest discretization rate in our experiment. Following the reviewer’s advice, we also set different discretization rates (presented as d, and "d=0.1" means that we remove 10% trajectory data regularly) in the SDD dataset and obtain some results in the following table.
>
> | Metric | d=0.1 | d=0.2 | d=0.3 | d=0.4 |
> |--|--|--|--|--|
> |ADE  | 8.24 | 12.21 | 15.03 | 19.49 |
> |FDE  | 16.74|19.58  | 24.98 | 30.87  |
>
> Table: Ablation study of discretization rate in SDD dataset.
>
> **Q5: Why do the authors not follow the experimental setup in [23] and also evaluate the ETH-UCY dataset?**
>
> A5: In Line 235, we have demonstrated that “We follow the [23] standard train-test split, and predict the future 4.8s”. Moreover, we have evaluated our method in the ETH-UCY dataset and the results are shown **in the supplementary material Table 6**. Clearly, this dataset can be labeled with actions and our method can obtain great performance in this dataset.
>
> **Q6: The organization of the experiments section is not convincing.**
>
> A6: Due to the page limitation, most of the experimental results are put into supplementary material, including ablation studies of backbones, additional results in the ETH/UCY dataset, etc. We will add more results in the main paper.
>
> **Q7: The authors claim their models have “superior interpretability” (L15) ...**
>
> A7: In Figure 5, we have added some explanation about the logic rule from the NBA dataset to show the superior interpretability of our method. Each rule can vividly manifest the corresponding actions of each player. Additionally, we also show more generated rules **in the attached PDF Table 1**. These rules also conform to the common sense. All equations will be numbered in the final version.
>
> **Q8: The Haggling dataset [a] contains triadic interactions which could have been utilized.**
>
> A8: We choose trajectory data because it’s easy to define spatial-temporal relation predicates and object property to explore some intrinsic logic rules. Moreover, following the reviewer’s advice, we also evaluate our method in the Haggling dataset, and the results are shown in the following table. Our method achieves promising results for both short-term and long-term predictions of complex activities. **More results are shown in the attached PDF Table 3 and Figure 1.**
>
> |Time|80ms|160ms|320ms|400ms|560ms|640ms|720ms|1000ms|
> |--|--|--|--|--|--|--|--|--|
> |Walking|0.38|0.58|0.80|0.89|0.98|1.03|1.11|1.22|
> |Eating|0.25|0.39|0.60|0.76|0.94|1.01|1.03|1.29|
>
> Table: Performance evaluation (in MAE) of comparison methods over on the Haggling (H3.6) dataset.
>
> **Q9: Does object mean humans?**
>
> A9: Generally, the object set C represents all objects in the specific environment. In our settings, all entities are human, so the “object” means human.
>
> **Q10: Could the authors elaborate on how k is encoded?**
>
> A10: Note that we have defined several event types k in Line 238 and Line 243. These actions can be encoded from the history of each entity as a booling logic variable.

---

> > ### Comment · Reviewer_zUPP · 2023-08-19
> >
> > Thank you for your response. The rebuttal as well as the other reviewers have addressed most of my concerns and I will change my recommendation to "accept".

---

> > > ### Author Response · Authors · 2023-08-20
> > > **Thank you!**
> > >
> > > Dear Reviewer zUPP:
> > >
> > > We would like to express our gratitude for your acknowledgment of our work! Do you mind updating my score accordingly in the openreview system? We'll incorporate the new results & comparison in the revised version!

---

### Official Review · Reviewer_gEM9 · 2023-07-06

**Soundness:** 3 good
**Presentation:** 3 good
**Contribution:** 3 good
**Rating:** 6
**Confidence:** 2

**Summary:**

This study presents a logic-informed, knowledge-driven modeling framework designed to predict and understand human movements, based on the analysis of their trajectories. It takes into account that human behaviors are commonly guided by intentions, desires, and spatial relationships with surrounding objects. The research integrates a set of spatial-temporal logic rules, derived from observational data, into the model, utilizing an expectation-maximization (EM) algorithm to infer model parameters and rule content. The performance of the model is evaluated using datasets of NBA basketball player movements and pedestrian trajectories, demonstrating high levels of interpretability, predictive accuracy, and efficacy in its results.

**Strengths:**

- The objectives and motivation are well-articulated, offering a clear understanding of the reasons behind the research and the problem it aims to address.
- The performance of the proposed model is outstanding when compared to other state-of-the-art methods signifying an advantage of the work.
- The proposed method is based on theoretical principles (EM method). It's not a product of arbitrary choices, but rather a thoughtful construction based on the EM algorithm, which adds credibility to the model's outcomes.

**Weaknesses:**

- If additional specifics regarding the architecture could be provided, such as the input and output formats utilized by the transformer, it would be incredibly beneficial for a deeper understanding of the method.
- The evaluation of this work is currently based on only two datasets. If the authors could expand this to include results from more universally employed pedestrian trajectory prediction datasets (ETH [i] or UCY [ii] ), it would significantly strengthen the claim of the model's generalizability.

  [i] Improving data association by joint modeling of pedestrian trajectories and groupings ECCV10.
  [ii] Crowds by example. In Computer Graphics Forum.

**Questions:**

- There is a typo in equation (3); the correct symbol should be $\varphi_f$ instead of $\phi_f$ .
- The references for the datasets utilized in the experiments are currently absent and need to be included.
- Where did you get the heatmap depicted in Fig3 (b) (d)?

**Limitations:**

If my understanding is correct, the spatial-temporal logic rule space, encompassing both static spatial and dynamic spatial relations, needs to be explicitly defined. This implies that prior knowledge or assumptions about the environment and the interactions within it are required to structure and define these rules. Thus, it's not a process automatically derived from the data but requires human intervention and understanding to set up appropriately.

---

> ### Author Rebuttal · Authors · 2023-08-08
>
> We sincerely thank you for your recognition of our work and your insightful reviews! We hope our responses can address your questions. Our responses are listed below.
>
> **Q1: If additional specifics regarding the architecture could be provided, such as the input and output formats utilized by the transformer, it would be incredibly beneficial for a deeper understanding of the method.**
>
> A1: Thanks for your suggestion. The input of our framework is the historical trajectory coordinates of entities, and the output is the predicted future trajectory. Coordinates as input would be first encoded into a vector by three-layer MLPs before being fed into the Transformer, and a ReLU nonlinearity following each of the first two layers. The dimensions of keys, values, and queries are all set to 256, and the hidden dimension of feed-forward layers is 512. The number of heads for multi-head attention is 8. We will add these details in a revision.
>
> **Q2: The evaluation of this work is currently based on only two datasets. If the authors could expand this to include results from more universally employed pedestrian trajectory prediction datasets (ETH [i] or UCY [ii] ), it would significantly strengthen the claim of the model's generalizability.**
>
> A2: We have evaluated our method and other state-of-the-art methods in the ETH/UCY dataset, and **due to the page limitation** in the main paper, the results are shown in **the supplementary material Table 6**. Below we show the results for the reviewer’s convenience. These new experimental results show that we still achieve superior results in most metrics.
>
> | Methods  | ETH | HOTEL | UNIV | ZARA1 | ZARA2 |AVG|
> |--|--|--|--|--|--|--|
> |Y-Net          |0.28/0.33|0.10/0.14|0.24/0.41|0.17/0.27|0.13/0.22|0.18/0.27|
> |MID            |0.39/0.66|0.13/0.22|0.22/0.45|0.17/0.30|0.13/0.27|0.21/0.38|
> |NSP-SFM        |0.25/0.44|0.09/0.13|0.21/0.38|0.16/0.27|0.12/0.20|0.17/0.24|
> |Social SSL     |0.69/1.37|0.24/0.44|0.51/0.93|0.42/0.84|0.34/0.67|0.44/0.85|
> |Social Implicit|0.66/1.44|0.20/0.36|0.32/0.60|0.25/0.50|0.22/0.43|0.33/0.37|
> |Social-VAE     |0.41/0.58|0.13/0.19|0.21/0.36|0.17/0.29|0.13/0.22|0.21/0.33|
> |ABC+           |0.31/0.44|0.16/0.21|0.25/0.47|0.21/0.28|0.20/0.26|0.23/0.32|
> |Ours           |**0.22/0.30**|**0.07/0.13**|**0.16/0.34**|**0.14/0.25**|**0.07/0.16**|**0.13/0.24**|
>
> Table: Quantitative results (ADE20/FDE20) of trajectory prediction in ETH/UCY dataset.
>
> **Q3: There is a typo in equation (3); The references for the datasets utilized in the experiments are currently absent and need to be included.**
>
> A3: These typos and references (SDD dataset [1] and ETH [2]/UCY [3] dataset) have been modified in a revision.
>
>
> [1] Robicquet A, Sadeghian A, Alahi A, et al. Learning social etiquette: Human trajectory understanding in crowded scenes[C]//Computer Vision–ECCV 2016: 14th European Conference, Amsterdam, The Netherlands, October 11-14, 2016, Proceedings, Part VIII 14. Springer International Publishing, 2016: 549-565.
>
> [2] Pellegrini S, Ess A, Schindler K, et al. You'll never walk alone: Modeling social behavior for multi-target tracking[C]//2009 IEEE 12th international conference on computer vision. IEEE, 2009: 261-268.
>
> [3] Lerner A, Chrysanthou Y, Lischinski D. Crowds by example[C]//Computer graphics forum. Oxford, UK: Blackwell Publishing Ltd, 2007, 26(3): 655-664.
>
> **Q4: Where did you get the heatmap depicted in Fig3 (b) (d)?**
>
> A4: Following [4][5], we utilize Fig. 3 to show the heatmaps of goals and waypoints for t=30 seconds in the future from the estimated distribution. These heat maps are achieved through an estimated distribution map obtained by the decoder in our framework.
>
> [4] Mangalam K, An Y, Girase H, et al. From goals, waypoints & paths to long term human trajectory forecasting[C]//Proceedings of the IEEE/CVF International Conference on Computer Vision. 2021: 15233-15242.
>
> [5] Jacobs H O, Hughes O K, Johnson-Roberson M, et al. Real-time certified probabilistic pedestrian forecasting[J]. IEEE Robotics and Automation Letters, 2017, 2(4): 2064-2071.

---

> > ### Comment · Reviewer_gEM9 · 2023-08-20
> >
> > Thank you for the comprehensive rebuttal. It effectively addresses my concerns. After considering the author's response and comments from other reviewers, I will maintain my initial rating.

---

### Author Rebuttal · Authors · 2023-08-09

Dear Reviewers, Area Chairs, and Program Chairs,

We are greatly thankful for the insightful comments and suggestions, which are very helpful for us to further improve this work. We are very excited that the reviewers hold positive feedback and find our work "well-articulated, offering a clear understanding....the performance of the proposed model is outstanding" (Reviewer gEM9), "the explainability of the model is impressive and well-visualized" (Reviewer zUPP), "introduces a novel framework that incorporates separate steps for rule generation and logic reasoning" (Reviewer 7WeJ), "the rule definition and rule inference framework is quite general and can find wider applications than the experiments" (Reviewer AEbP). In our response, we provide additional figures and tables in the attached PDF for visualization. Recognizing the importance of additional explanations and experiments, we are committed to making these enhancements. To ensure transparency and clear communication, we've summarized our main responses as follows:

- **Experiment on ETH/UCY dataset.** We would like to address the fact that some reviewers may not have had the opportunity to review our supplementary material, leading to concerns regarding the ETH/UCY dataset. It's worth noting that we have indeed evaluated our method alongside other state-of-the-art techniques using the ETH/UCY dataset. However, owing to page constraints in the main paper, we've presented the results in *Table 6 of the supplementary material*.

- **Experiment on 3D scenarios.** In response to the reviewer's suggestion, we have extended our evaluation to the 3D Haggling dataset, encompassing diverse action types. The outcomes are detailed in *Table 3* and *Figure 1* in the attached PDF. Encouragingly, our method demonstrates noteworthy performance for both short-term and long-term predictions, particularly in intricate activity scenarios.

- **Experiment on synthetic datasets.** We use the approach outlined in [1] to test our model's ability to uncover rules on synthetic datasets with known ground-truth rules and weights. We compare the learned rule weights with the actual ones and report the Mean Absolute Error (MAE) as a measure of accuracy.

- **Ablation study.** Based on the feedback from reviewers, we have included more ablation studies in our work. These studies cover aspects such as rule selection, discretization rates, and variations in different backbones.

- **Clarification about some important concepts.** We have revised our explanation of "logic" and "interpretability" based on the comments provided by the reviewers.

Beyond the academic contributions presented in the paper, our approach also offers practical significance. We introduce a feasible and differentiable algorithm capable of simultaneous learning of rule content and model parameters from observational data. Our model directly utilizes precise, detailed, and irregularly-spaced action times and original 3D coordinates as inputs. Through extensive experimentation on real datasets, we have showcased our model's strong performance in human action prediction and explanation. We hold the view that this distinct approach has the potential to inspire a host of future research endeavors.

[1] Li S, Feng M, Wang L, et al. Explaining point processes by learning interpretable temporal logic rules[C]//International Conference on Learning Representations. 2021.

---

### Decision · Program_Chairs · 2023-09-21

**Decision:**

Accept (poster)

**Comment:**

This paper presents an interesting framework for discovering rules for action recognition in a spatial-temporal logic setup. The methodological contribution includes a well-formulated EM approach for learning logic rules to explain a dataset of actions. The empirical results are also convincing, demonstrating the efficacy of the approach.

In the rebuttal/discussion, the authors clarified details, terminology, and pointed to synthetic/real dataset results that support the claims in the paper. The reviewers concur on an accept rating for the paper.